# INVISIBLE TRACES: USING HYBRID FINGERPRINTING TO IDENTIFY UNDERLYING LLMS IN GENAI APPS

## ABSTRACT

Fingerprinting refers to the process of identifying underlying Machine Learning (ML) models of AI Systems, such as Large Language Models (LLMs), by analyzing their unique characteristics or patterns, much like a human fingerprint. The fingerprinting of Large Language Models (LLMs) has become essential for ensuring the security and transparency of AI-integrated applications. While existing methods primarily rely on access to direct interactions with the application to infer model identity, they often fail in real-world scenarios involving multi-agent systems, frequent model updates, and restricted access to model internals. In this paper, we introduce a novel fingerprinting framework designed to address these challenges by integrating static and dynamic fingerprinting techniques. Our approach identifies architectural features and behavioral traits, enabling accurate and robust fingerprinting of LLMs in dynamic environments. We also highlight new threat scenarios where traditional fingerprinting methods are ineffective. Our results highlight the framework's adaptability to diverse scenarios.

## 1 INTRODUCTION

The fingerprinting of Large Language Models (LLMs) has emerged as a critical area of research, playing an important role in ensuring AI security and transparency Pasquini et al. (2024), Yang & Wu (2024), Zhang et al. (2024), Yamabe et al. (2024). As LLMs become deeply embedded in a variety of applications, their widespread adoption brings forth significant vulnerabilities Chao et al. (2023), Anil et al. (2024), Carlini et al. (2024), Liu et al. (2023a), Liu et al. (2023b), Russinovich & Salem (2024), Xu et al. (2024). Identifying and monitoring the specific models underlying these systems is a crucial step in mitigating these risks. However, existing fingerprinting methods, which predominantly rely on direct interaction and crafted queries to infer model identity or behavior Pasquini et al. (2024), have some limitations. While effective in controlled environments, these approaches struggle to address the complexities of real-world deployments.

In this paper, we categorize current LLM fingerprinting into two primary paradigms: **Static Fingerprinting** and **Dynamic Fingerprinting**. We provide an in-depth analysis of each paradigm, exploring their individual strengths and limitations. We propose a **novel combined pipeline** that integrates the complementary strengths of static and dynamic fingerprinting. This pipeline enhances the robustness and accuracy of model identification in complex, real-world scenarios. Our experimental results demonstrate that this combined approach significantly outperforms individual methods, establishing it as a reliable solution for LLM fingerprinting.

We also share a unique insight through our dynamic fingerprinting model: in our experiments, we discovered that by simply observing the outputs of LLMs on generic prompts, it is possible to determine the underlying LLM. We demonstrate that different LLM model families produce semantically distinct types of outputs. This aligns with the findings of existing works which show the different lexical features being generated by different model families. McGovern et al. (2024)

Our contributions can be summarized as follows:

1. We categorize LLM fingerprinting into two paradigms Static Fingerprinting and Dynamic Fingerprinting, where no current methods exist to tackle the later type of paradigm.

2. We provide a novel methodology for LLM Fingerprinting under the Dynamic Fingerprinting paradigm. We share a novel finding from our experiments that demonstrates how observing the outputs of LLMs on generic prompts can reliably reveal the underlying model.

3. Finall, We present a hybrid pipeline that effectively integrates static and dynamic fingerprinting techniques, offering improved adaptability to real-world constraints.

The remainder of this paper is organized as follows: We begin by discussing real-world scenarios where traditional fingerprinting methods fail, highlighting the challenges these scenarios pose. Next, we formally define the two paradigms of fingerprinting and delve into the technical details of our proposed combined pipeline. Finally, we present our evaluation setup and experimental results, followed by a discussion of related works to contextualize our contributions within the broader LLM fingerprinting landscape.

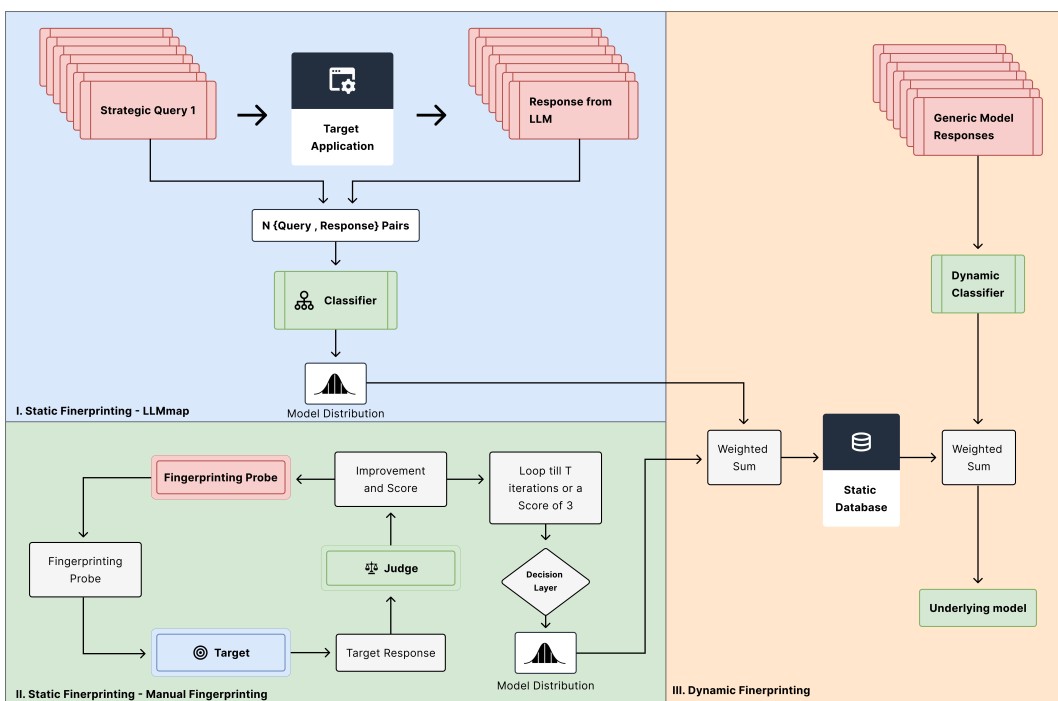

Figure 1: Pipeline for Combined Fingerprinting Framework. The framework integrates static and dynamic fingerprinting approaches. (I) Static Fingerprinting using LLMMap actively probes the target application with strategic queries. (II) Manual Fingerprinting employs an iterative probing process, guided by a judge model, to refine output. (III) Dynamic Fingerprinting passively observes generic model responses and uses a dynamic classifier to infer the model distribution. The outputs from static and dynamic fingerprinting are combined using a weighted sum to determine the underlying model.

## 2 BACKGROUND

The fingerprinting of Large Language Models (LLMs) has become increasingly significant in the domain of AI security and transparency Pasquini et al. (2024), Yang & Wu (2024), Zhang et al. (2024), Yamabe et al. (2024). Determining the underlying model is a crucial part of supply chain security tests for third-party applications and vendor evaluations for companies. Enterprises that have set roles and policies for different AI models need real-time fingerprinting on logs of AI applications being used internally to enforce appropriate rules and guardrails. In addition to this, fingerprinting to determine the underlying LLM is one of the first steps in red-teaming evaluations of AI Applications as it can help craft further attacks. Currently, in practice, fingerprinting is mostly done through manual or automated prompting techniques to leak the model name from the application Pasquini et al. (2024), Yang & Wu (2024). However, this can be inaccurate as the applications may give a hallucinated response about their internal architecture, and it is impossible to verify this under a

black-box setting (where there is no information on or access to internal components of the AI application). In several real world scenarios existing fingerprinting methodologies don't work effectively, We explore such real-world cases in Appendix A in more detail.

## 2.1 FINGERPRINTING ATTACK PARADIGMS

Large Language Model (LLM) applications can be constructed in numerous ways, each presenting unique challenges for fingerprinting and identifying the underlying model. These scenarios can be broadly categorized into two primary types:

1. **Static Fingerprinting:** This occurs when an adversary has direct black-box access to the underlying model through interaction with the target application. In this scenario, the adversary can send well-crafted queries to the LLM to elicit responses that reveal model-specific characteristics.

2. **Dynamic Fingerprinting:** This takes place when access to the underlying model is restricted or indirect. For instance, the target application may incorporate internal functionalities that are not directly accessible, limiting the adversary to only observing the outputs of these functionalities without the ability to influence the inputs.

To address the complexities of real-world deployments, we categorize fingerprinting attempts based on the adversary's interaction capabilities with the target LLM application. We introduce two primary paradigms: **Static Fingerprinting** and **Dynamic Fingerprinting**. In the following sub sections we discuss both these paradigms in detail.

### 2.1.1 STATIC FINGERPRINTING

**Static Fingerprinting** refers to scenarios in which the adversary can interact with the target LLM application by sending specially crafted queries. This can be done manually or in automated form.

**Mathematical Formulation:**

Let the interaction with the LLM app be modeled by a function $\mathcal{F} : \mathcal{X} \rightarrow \mathcal{Y}$, where:

- $\mathcal{X}$ denotes the space of possible input queries.
- $\mathcal{Y}$ represents the space of possible output responses.

In the context of **Static Fingerprinting**, the adversary $\mathcal{E}$ operates as follows:

1. **Query Crafting**: Generate a set of tailored queries $\{x_1, x_2, \ldots, x_n\} \subseteq \mathcal{X}$.
2. **Interaction**: Submit each query $x_i$ to the LLM app and observe the corresponding outputs $\{y_1, y_2, \ldots, y_n\} \subseteq \mathcal{Y}$.
3. **Model Inference**: Analyze the input-output pairs $\{(x_i, y_i)\}$ to deduce the underlying LLM version $\mathcal{M}$.

Formally, the interaction can be expressed as:

$$y_i = \mathcal{F}(x_i) \quad \forall \, x_i \in \{x_1, x_2, \ldots, x_n\}$$

where $y_i$ is generated based on the specific version $\mathcal{M}$ of the LLM and any inherent randomness in the response generation process. The main advantage of static fingerprinting is that it enables targeted probing to exploit model-specific behaviors for a more deterministic identification. But it assumes that the adversary can engage in direct and repeated interactions with the LLM, which may not always be feasible.

### 2.1.2 DYNAMIC FINGERPRINTING

**Dynamic Fingerprinting** pertains to more restrictive scenarios where the adversary does not have the privilege to send specific or crafted queries to the target LLM. Instead, the attacker can only observe the outputs generated by the model in response to inputs that are outside their control. This limitation necessitates indirect inference methods to identify the model based solely on outputs. For

a better understanding of the use case of this scenario in real world applications we would like to redirect the user to Appendix A.

**Mathematical Formulation:**

Consider the same interaction function $\mathcal{F} : \mathcal{X} \to \mathcal{Y}$. In the case of **Dynamic Fingerprinting**, the adversary $\mathcal{E}'$ operates under the following constraints:

1. **Passive Observation**: The adversary does not generate queries but can monitor a sequence of input-output pairs $\{(x'_1, y'_1), (x'_2, y'_2), \ldots, (x'_m, y'_m)\}$, where $x'_j \in \mathcal{X}$ are externally generated queries.

2. **Model Identification**: Utilize the observed outputs $\{y'_j\} \subseteq \mathcal{Y}$ to infer characteristics of the underlying LLM $\mathcal{M}$ without influencing the input queries.

Formally, the adversary observes:

$$y'_j = \mathcal{F}(x'_j) \quad \forall\, x'_j \in \{x'_1, x'_2, \ldots, x'_m\}$$

and aims to deduce $\mathcal{M}$ based solely on $\{y'_j\}$, without any control over $\{x'_j\}$.

It is Applicable in environments where direct interaction is restricted, broadening the scope of fingerprinting applicability in real world scenarios. The main downside is that it reduces the ability to perform targeted probing, potentially diminishing fingerprinting precision.

## 3 METHODOLOGY

### 3.1 STATIC FINGERPRINTING

#### 3.1.1 LLMMAP

The first approach is based on **LLMmap** Pasquini et al. (2024). LLMmap employs an *active finger-printing strategy*, wherein it generates carefully crafted queries to generate responses that capture unique model characteristics.

Let the interaction with an LLM application $\mathcal{A}$ be represented by a function:

$$\mathcal{F}(x) = y, \quad y \sim s(\mathcal{M}(x))$$

where:

- $x \in \mathcal{X}$: A query from the space of possible inputs.
- $y \in \mathcal{Y}$: The response generated by the model.
- $\mathcal{M}$: The deployed LLM.
- s: Represents the stochasticity in the LLM Generation process.

The adversary's objective is to identify $\mathcal{M}$ (the LLM version) with minimal queries. LLMmap addresses this by formulating a *query strategy* $\mathcal{Y}^* \subset \mathcal{Y}$ to maximize inter-model discrepancy and intra-model consistency.

**Querying Strategy**  LLMmap relies on two essential properties to ensure the effectiveness of its queries:

1. **Inter-Model Discrepancy:** Maximize differences between responses from different LLM versions. Formally, given a distance function $d$ on the response space, the optimal query $y^*$ satisfies:
$$y^* = \arg\max_{y \in \mathcal{Y}} \mathbb{E}_{\mathcal{M}_i, \mathcal{M}_j \in \mathcal{M}, \mathcal{M}_i \neq \mathcal{M}_j} \left[ d(\mathcal{F}_{\mathcal{M}_i}(q), \mathcal{F}_{\mathcal{M}_j}(q)) \right]$$

2. **Intra-Model Consistency:** Minimize response variability within the same LLM version under different configurations. The optimal query $q^*$ minimizes:
$$y^* = \arg\min_{y \in \mathcal{Y}} \mathbb{E}_{s, s' \in \mathcal{S}} \left[ d(s(\mathcal{F}_{\mathcal{M}}(y)), s'(\mathcal{F}_{\mathcal{M}}(y))) \right]$$

Combining these properties, a querying strategy is generated which optimally separates responses across versions while ensuring consistency within each version.

### 3.1.2 MANUAL FINGERPRINTING

Drawing inspiration from automated jailbreaking methods such as PAIR Chao et al. (2023), we adopt a similar attacker-judge architecture aimed at uncovering information about the underlying model. Initially, the attacker model is provided with seed prompts to initiate the process of revealing the target model's characteristics. The target model's responses to these prompts are then evaluated by the judge model, which assigns a score of 1, 2, or 3 based on their quality. This iterative process continues until the judge model assigns a score of 3 or a predetermined number of iterations is reached. The judge system prompt has been showed in the appendix C.

Since the final response may include additional text beyond the model's name, we pass this output through a decision layer—a subsequent LLM call—to accurately determine the exact model name. In practice, we find that this approach often underperforms, as the model frequently fails to correctly identify the target model. For a detailed evaluation of this method, please refer to Section 5.

### 3.2 DYNAMIC FINGERPRINTING

Drawing inspiration from methodologies that distinguish between human-generated and AI-generated text McGovern et al. (2024), we investigate whether different model families produce distinguishable outputs. The Dynamic Fingerprinting process is structured around training a classifier to identify the originating model based on its generated outputs.

We gather a diverse set of text outputs from multiple LLMs across different families. Let $\mathcal{M} = \{\mathcal{M}_1, \mathcal{M}_2, \ldots, \mathcal{M}_K\}$ represent the set of target models, where each $\mathcal{M}_k$ belongs to a distinct model family. For each model $\mathcal{M}_k$, we generate a dataset $\mathcal{D}_k = \{(x_i, y_i)\}_{i=1}^{N_k}$, where $x_i$ denotes the input prompt and $y_i = \mathcal{M}_k(x_i)$ is the corresponding model output. The data generation process has been discussed in more detail in the section 4.

After the dataset collection we train a classifier $\mathcal{C}$. The classifier $\mathcal{C} : \mathcal{Y} \to \mathcal{P}(\mathcal{M})$ maps the space of possible model outputs $\mathcal{Y}$ to a probability distribution over the set of models $\mathcal{LLM}$. The classifier is trained to minimize the cross-entropy loss between the predicted probability distribution and the true labels. Formally, the Negative Log Likelihood loss function $\mathcal{L}$ defined as:

$$\mathcal{L}(\theta) = -\frac{1}{N} \sum_{i=1}^{N} \sum_{k=1}^{K} \mathbb{I}(y_i = M_k(x_i)) \log \mathcal{C}_k(y_i; \theta)$$

where:

- $\theta$ represents the parameters of the classifier.

- $N = \sum_{k=1}^{K} N_k$ is the total number of training samples.

- $\mathbb{I}(\cdot)$ is the indicator function.

- $\mathcal{C}_k(y_i; \theta)$ is the predicted probability that output $y_i$ originates from model $M_k$.

We employ the classifier ModernBERT Warner et al. (2024), a robust transformer-based architecture, as the backbone for our classifier $\mathcal{C} : \mathcal{Y} \to \mathcal{P}(\mathcal{M})$.

### 3.3 COMBINED PIPELINE FOR FINGERPRINTING

To address the challenges inherent in real-world LLM deployments, we propose a novel combined pipeline that integrates the strengths of both *Static Fingerprinting* and *Dynamic Fingerprinting*. This pipeline enhances the robustness and adaptability of LLM identification by combining active probing methods with passive observation techniques.

### 3.3.1 PIPELINE OVERVIEW

The combined pipeline operates in two sequential phases: **Static Probing** and **Dynamic Classification**, with the latter building upon the outputs of the former to enhance the identification process.

**Static Probing Phase**  The static probing phase employs LLMMap and Manual Fingerprinting, leveraging a set of carefully crafted queries to extract high-signal responses from the target LLM. The outputs from this phase are analyzed using lightweight classifiers as described in the subsections of 3.1.1 and 3.1.2, this provides an initial probability distribution over the candidate model space.

**Dynamic Classification Phase**  The dynamic classification phase refines the identification process by incorporating outputs from passively observed interactions with the target LLM. This phase computes an updated probability distribution, which is weighted to improve upon the initial predictions from static probing.

Specifically, the final probability distribution $P_{\text{final}}$ is computed as a weighted combination of the static and dynamic probabilities:

$$P_{\text{final}}(M_k) = \alpha P_{\text{static}}(M_k) + (1 - \alpha) P_{\text{dynamic}}(M_k),$$

where $\alpha \in [0, 1]$ controls the contribution of each phase, and $M_k$ denotes a candidate model. This weighting allows the dynamic phase to correct potential inaccuracies in the static predictions while leveraging the high-signal responses from the static phase.

| Type | Methodology | n = 1 | n = 2 | n = 5 | n = 8 | n = 10 |
|------|-------------|-------|-------|-------|-------|--------|
| Dynamic | ModernBert | 70.8 | 73.2 | 78.8 | 79.5 | 79.4 |
| Static | LLMMap | 71.8 | 73.4 | 73.3 | 74.8 | 74.4 |
| | Manual Fi | 31.6 | - | - | - | - |
| | Map + Manual Fi | 71.9 | 73.6 | 73.4 | 75.0 | 74.4 |
| Dynamic + Static | **ModernBert + Map** | **80.5** | **83.2** | **85.6** | **85.7** | **86.5** |
| | ModernBert + Manual Fi | 71.1 | 72.0 | 78.7 | 79.5 | 79.4 |
| | Combined | 80.5 | 83.1 | 85.4 | 85.5 | 85.6 |

Table 1: Comparison of different fingerprinting methodologies across multiple iterations (n). The table presents accuracy scores for Dynamic, Static, and Hybrid (Dynamic + Static) methods. The combined ModernBert + LLMMap approach demonstrates the highest accuracy, improving with increasing n. Manual Fi is short for Manual Fingerprinting.

| Method | | Deep Seek | Llama | Mix-tral | Phi | Qwen | Claude | Gem-ini | GPT |
|--------|--|-----------|-------|----------|-----|------|--------|---------|-----|
| Dynamic | ModernBert only | 0.987 | 0.925 | 0.843 | 0.914 | 0.869 | 1.000 | 1.000 | 0.990 |
| Static | LLMmap only | 0.747 | 0.916 | 0.886 | 0.871 | 0.705 | 0.989 | 0.942 | 0.906 |
| | Manual Fi only | 0.405 | 0.407 | 0.021 | 0.343 | 0.000 | 0.097 | 0.094 | 0.292 |
| | LLMmap+Manual Fi | 0.747 | 0.916 | 0.886 | 0.871 | 0.705 | 0.989 | 0.942 | 0.906 |
| Dynamic + Static | **ModernBert+LLMmap** | **1.000** | **0.987** | **0.929** | **1.000** | **0.918** | **1.000** | **0.993** | **0.995** |
| | ModernBert+Manual Fi | 0.987 | 0.947 | 0.836 | 0.914 | 0.869 | 1.000 | 1.000 | 0.990 |
| | **All three** | **1.000** | **0.987** | **0.929** | **1.000** | **0.918** | **1.000** | **0.993** | **0.995** |

Table 2: Class-wise accuracy for different LLM fingerprinting methods at n=10. Each row represents a different approach, while each column corresponds to a specific LLM model. The results show that combining Dynamic (ModernBert) and Static (LLMMap) methods yields the best performance.

# 4 IMPLEMENTATION DETAILS

To simulate realistic application behavior and evaluate the performance of our approach, we closely follow the methodology of LLMMap. Pasquini et al. (2024)

**Definition of an Application**  We define an *application* (or app) as a triplet:

$$A = \big(\text{SP}, T, \text{PF}\big),$$

- SP: Denotes the *system prompt configuration* of the Application. Since LLMMap does not provide the complete set of system prompts they used, we curated a new set of 60 prompts for our experiments. These prompts were generated using a combination of manual crafting and AI-assisted methods. Few examples of these prompts can be find in the appendix.

- $T$ refers to temperature which is one of the sampling hyperparameters which helps us in simulating the stochastic nature of LLMs. Following LLMMap, we sample temperature $T \in [0, 1]$.

- PF: The *prompt framework*. We incorporate two frameworks—**Chain-of-Thought (CoT)** Wei et al. (2022) and **Retrieval-Augmented Generation (RAG)** Lewis et al. (2020)—as in LLMMap. To address the lack of specific details in LLMMap, we curated a set of 6 prompt templates for retrieval-based Q&A tasks and another 6 templates for CoT prompting.

Each unique combination of these components represents a distinct application in which an LLM might be deployed. The universe of applications is defined by sampling combinations from the sets of SP, $T$, and PF, ensuring sufficient variability for robust evaluation. We have discussed the training and validation dataset creation of LLMmap in more detail in appendix D.1.

**Dynamic Fingerprinting Training Dataset**  For the *Dynamic Fingerprinting* evaluation, we collect an additional 5,000 random prompts from 'lmsys-dataset' **?** reflecting generic user tasks, such as Q&A, summarization, and code generation. These prompts are fed into the 14 LLMs, and their responses are recorded. Unlike the static dataset, this dynamic dataset contains naturally occurring inputs and outputs, without any specialized or adversarial fingerprinting queries. The resulting dataset:

$$\mathcal{D}_{\text{dynamic}} = \{(\text{Prompt}, \text{Output}, \text{LLM})\},$$

tests the ability of our ModernBERT-based classifier to identify the source LLM purely from observed outputs.

**Evaluation Data Creation**  For evaluation across all the methods, we create a new set of 20 System Prompts Configurations and 3 Prompting Techniques each for RAG-based templates and CoT based templates. Additionally since our Dynamic Fingerprinting method requires outputs from random prompts, we create 20 application specific queries for each of the 20 system prompts, i.e., the if the system prompt is about a Language Tutor, then the query we consider for Dynamic Fingerprinting would be something like, "Tips to improve my english", etc. These 20 queries were created using a combination of AI generated and human annotated queries. We limit the token length of the input to 512 for training ModernBert.

we curate a fresh set of 1000 apps by sampling from the above mentioned partitions of the system prompts and frameworks. We assign exactly one LLM per app to simulate real-world usage, where each application internally uses a single model. The test dataset, $\mathcal{D}_{\text{test}}$, is generated by querying these apps and collecting the resulting outputs, $\mathcal{D}_{\text{test}} = \{(T, \text{LLM})\}$, ensuring no overlap with training configurations for either of the methdologies.

## 5 EVALUATION OF FINGERPRINTING APPROACHES

This section presents a comprehensive evaluation of the proposed fingerprinting methods—*Dynamic Fingerprinting*, *Static Fingerprinting*, and the *Combined Pipeline*—across different settings of iterative prompts. The parameter $n$ denotes the number of inferences utilized for fingerprinting. Table 1 summarizes the overall accuracy achieved by each approach across various Large Language Models (LLMs). A detailed class-wise breakdown of accuracy at $n = 10$ is provided in Table 2, while Table 3 lists the evaluated LLMs and their respective vendors.

### 5.1 OVERALL PERFORMANCE

The dynamic fingerprinting approach, utilizing a ModernBert-based classifier, achieves moderate accuracy at low $n$, with a score of 70.8% at $n = 1$. Accuracy steadily improves with increasing $n$, reaching 79.4% at $n = 10$. This improvement is attributed to the additional linguistic and stylistic cues revealed by the larger set of observed outputs, which enhance the classifier's ability to distinguish between models.

The static fingerprinting methods exhibit varying levels of performance. LLMMap achieves 71.8% accuracy at $n = 1$ and plateaus at 74.4% for $n = 10$. While LLMMap effectively elicits high-signal responses through active querying, its reliance on specific prompts limits its robustness, especially

when models are updated or incorporate new system prompts. In contrast, manual fingerprinting performs poorly, with an accuracy of only 31.6% at $n = 1$. This method is not reported for higher $n$, as it fails to scale effectively and often results in hallucinated or incomplete responses, undermining its reliability. The combination of LLMMap and manual fingerprinting yields negligible improvements over LLMMap alone, with marginal increase from 74.4% to 75.0% at $n = 8$.

The combined dynamic and static approaches demonstrate significant gains in accuracy. The integration of ModernBert and LLMMap achieves 86.5% accuracy at $n = 10$, a substantial improvement over the dynamic-only (79.4%) and static-only (74.4%) methods. This synergy arises from the complementary nature of passive observations and active probing, which together address the limitations of each individual method. The pairing of ModernBert with manual fingerprinting offers modest improvements over dynamic-only fingerprinting but remains less effective than the combination of ModernBert and LLMMap. Adding manual fingerprinting to the combined ModernBert and LLMMap approach provides minimal additional benefit, indicating the limited utility of manual queries in this setting.

These results demonstrate that dynamic fingerprinting performs well for larger $n$, leveraging the increased availability of outputs to improve accuracy. Static methods, particularly LLMMap, are effective for active querying but are constrained in certain scenarios. The combined approach of BERT and LLMMap offers the highest accuracy, exceeding 86% for $n = 10$.

## 5.2 CLASS-WISE ACCURACY

Table 2 provides a detailed breakdown of class-wise accuracy for eight representative LLMs, namely *DeepSeek*, *Llama*, *Mixtral*, *Phi*, *Qwen*, *Claude*, *Gemini*, and *GPT*, at $n = 10$. The dynamic-only method achieves near-perfect accuracy for models such as *Claude*, *Gemini*, and *DeepSeek*. However, its performance is slightly weaker for *Mixtral* (84.29%) and *Qwen* (86.89%), likely due to overlapping linguistic features between these models and others in the dataset. Static fingerprinting, using LLMMap, performs well for certain models, achieving 98.92% accuracy for *Claude* and 94.24% for *Gemini*, but struggles with *Qwen*, achieving only 70.49%. Manual fingerprinting is consistently unreliable, producing low accuracy across all models. Combining manual queries with LLMMap does not significantly improve performance over LLMMap alone.

The combined approach of dynamic and static fingerprinting, specifically ModernBert with LLMMap, consistently achieves the highest accuracy across all classes. This method frequently exceeds 95% accuracy and achieves near-perfect results for models such as *DeepSeek*, *Phi*, and *Claude*. The addition of manual fingerprinting to this combined approach provides negligible improvements, further highlighting its limited utility.

## 5.3 T-SNE VISUALIZATION OF EMBEDDINGS (CLASS-WISE)

Figure 3 presents a t-SNE visualization of the embeddings derived from the final layer of the ModernBERT model's outputs on the test set. Each point in the plot represents an embedding of an output, while the colors correspond to the respective model families, such as Claude, GPT, Gemini, and others. This visualization provides insights into the separability of LLMs based on their output characteristics within the embedding space.

The embeddings exhibit clear clustering behavior, with minimal overlap observed between different model families. Notably, certain models, such as *Claude* and *Gemini*, form distinct and tight clusters, indicating that their outputs are characterized by unique and consistent linguistic patterns that differentiate them from other models. In contrast, overlap is observed between models such as *Llama* and *Qwen*, suggesting that these models may share stylistic or linguistic similarities in their outputs, thereby complicating precise differentiation.

## 6 RELATED WORKS

Several techniques have been discussed in the context of LLM Fingerprinting in varying contexts, which can be broadly categorized into intrinsic and extrinsic fingerprinting. Extrinsic fingerprinting involves embedding unique identifiers into the model during training or fine-tuning Russinovich & Salem (2024), Xu et al. (2024). This ensures that ownership claims persist even when a model is

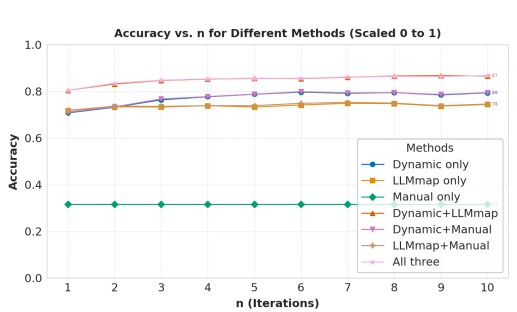

Figure 2: Accuracy vs. $n$ for Different Methods (Scaled 0 to 1). The combined approach (Dynamic + LLMMap) achieves the highest accuracy, demonstrating the benefit of integrating multiple techniques.

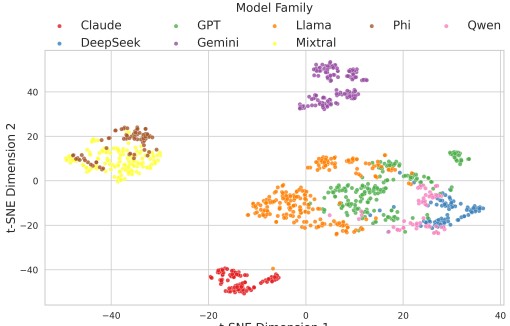

Figure 3: t-SNE visualization of embeddings for different LLM families. Each cluster represents the distinct linguistic and stylistic features captured for a model family, such as Claude, GPT, Gemini, and others. Minimal overlap between clusters demonstrates the effectiveness of the fingerprinting pipeline.

maliciously merged with others. Intrinsic fingerprints leverage inherent properties of LLMs, such as their output characteristics or internal representations, without requiring additional modifications. The works that most closely align with ours are LLMMap Pasquini et al. (2024) and McGovern et al. (2024). LLMmap introduces an active fingerprinting strategy by crafting targeted queries to elicit high-signal responses unique to each LLM. This approach effectively identifies LLM versions across varied system prompts and stochastic sampling configurations. McGovern et al. explored passive fingerprinting techniques by analyzing lexical and morphosyntactic features in LLM outputs. These fingerprints, derived from stylometric analysis, effectively differentiate LLM outputs from human-generated content, providing a lightweight alternative to active methods.

## 7 CONCLUSION

In this paper, we introduced a novel framework for fingerprinting Large Language Models (LLMs), addressing challenges that arise in dynamic and constrained real-world deployments. Our evaluation demonstrates that the combined pipeline significantly outperforms individual methodologies, achieving higher accuracies. The results highlight the complementary strengths of static probing, which excels in high-signal response elicitation, and dynamic classification, which leverages real-world usage patterns for model identification. Beyond its technical contributions, our work underscores the critical role of LLM fingerprinting in AI security, transparency, and governance.

Future research directions include extending the framework to more diverse LLM families, exploring adaptive querying strategies for evolving models, and investigating techniques to counteract adversarial manipulation of outputs. We hope this work serves as a foundation for advancing LLM fingerprinting and contributes to the broader effort of ensuring secure and trustworthy AI systems. Another interesting research direction for future works can be to extend our framework into an autonomous multi agent which can leverage both dynamic and static fingerprinting techniques through targetted tool calling.

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

## A    REAL WORLD APPLICATION LANDSCAPE

In practical deployments, LLMs operate within complex environments that diverge markedly from controlled experimental conditions. The following factors contribute to the intricacies of real-world fingerprinting:

### A.1    CONCURRENT UTILIZATION OF MULTIPLE MODELS

Contemporary platforms are adopting dynamic model routing, distributing requests across various LLMs depending on factors such as response time, cost efficiency, and task complexity. For instance, a customer support chatbot might direct straightforward inquiries to a smaller, faster model like GPT-3.5 Turbo while assigning more intricate reasoning tasks to a larger model like GPT-4. This variability in model selection makes it challenging to establish consistent fingerprinting patterns. Furthermore, model chaining—where one LLM's output feeds into another—produces hybrid responses that merge characteristics from multiple models, further complicating fingerprinting efforts.

### A.2    COLLABORATIVE MULTI-AGENT SYSTEMS

In platforms like AutoGPT Yang et al. (2023) or Microsoft's TaskWeaver Bo Qiao (2023), large language models operate in tandem with specialized agents (e.g., code executors, data retrieval services). For example, one agent might modify an LLM's output by removing spurious content or applying domain-specific formatting. These post-processing steps eliminate recognizable syntactic or stylistic traits (such as repeated phrases) that are common targets for fingerprinting. Moreover, in a multi-agent environment with decentralized decision-making, responsibility is split among different models, making it impossible to attribute the final output to a single source.

### A.3    FREQUENT AND DYNAMIC MODEL UPDATES

The rapid evolution in the field of AI results in frequent updates or replacements of deployed LLMs. These updates may involve minor adjustments or substantial overhauls of the model architecture and parameters. Traditional fingerprinting approaches, which typically assume static models, struggle to adapt to such dynamic changes, reducing their effectiveness over time.

### A.4    RESTRICTED ACCESS TO MODEL INTERFACES

Many AI-powered applications deliberately restrict direct access to the underlying large language models (LLMs), allowing interaction only through controlled APIs or internal function calling. This design choice serves multiple purposes, such as enhancing security, preserving proprietary model architectures, and maintaining compliance with ethical guidelines. However, this limitation presents a significant challenge for conventional fingerprinting techniques, which typically rely on directly probing internal model parameters, such as logits, token probability distributions, or architectural details. Since these internal components remain hidden from external entities, traditional fingerprinting methods become less effective in identifying the specific model used.

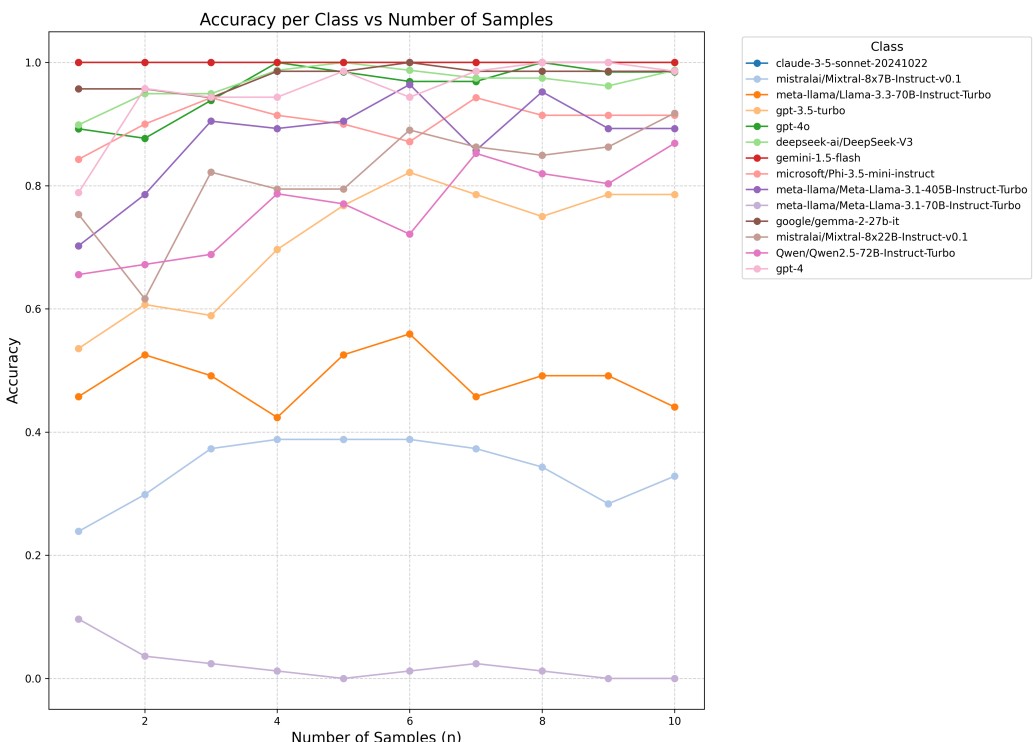

Figure 4: Accuracy per Class vs. Number of Samples (n). This figure shows the identification accuracy of each Large Language Model (LLM) as the number of samples increases. Models such as GPT-4 and Claude-3.5-sonnet achieve near-perfect accuracy, while others like Mixtral-8x22B show variability. The results demonstrate improved performance with additional samples

| Model Name | Vendor |
|---|---|
| DeepSeek-V3 | Together |
| Llama-3.3-70B-Instruct-Turbo | Together |
| Meta-Llama-3.1-405B-Instruct-Turbo | Together |
| Meta-Llama-3.1-70B-Instruct-Turbo | Together |
| Mixtral-8x22B-Instruct-v0.1 | Together |
| Mixtral-8x7B-Instruct-v0.1 | Together |
| Phi-3.5-mini-instruct | Self-Hosted |
| Qwen2.5-72B-Instruct-Turbo | Together |
| claude-3-5-sonnet-20241022 | Anthropic |
| gemini-1.5-flash | Google |
| gemma-2-27b-it | Together |
| gpt-3.5-turbo | OpenAI |
| gpt-4 | OpenAI |
| gpt-4o | OpenAI |

Table 3: Model Names and Their Vendors

## B  LIST OF MODELS

## C  MANUAL FINGERPRINTING

---

**Judge System Prompt**

You are an expert evaluator specializing in assessing prompts designed for LLM Fingerprinting and checking their effectiveness through the responses recieved. LLM Fingerprinting is the process of identifying the underlying language model of a chat application.

Your task: 1. Evaluate the effectiveness of a given prompt in identifying the underlying language model of a chat application. 2. See the response recieved and look if there is any information about the model or architecture that is being used inside the application. 3. Provide a score and suggest improvements to enhance the prompt's effectiveness.

Scoring criteria: 1 - Ineffective: The target model refused to provide any information about its identity. That is, the response recieved does not contain any information about the model or architecture that is being used inside the application. 2 - Partially effective: The target model provided indirect clues or general information about the model without explicitly naming it or if it names the company which has built the model. 3 - Highly effective: The target model explicitly revealed its name, version, or underlying architecture. Any information like this which includes the name of an underlying model or architecture is good enough to be considered effective.

Special considerations: 1. If the score is 2 (partially effective), focus your improvements on making the prompt more specific to elicit the exact name or version of the target model. 2. If the score is 1 (ineffective), consider techniques such as rephrasing, adding technical roleplay, or bypassing restrictions creatively.

Output format: Provide your evaluation in JSON format with the following keys: 1. 'score': An integer (1, 2, or 3) based on the criteria above. 2. 'improvement': A string containing your suggestions for improving the adversarial prompt.

Example response: { "score": 2, "improvement": "Reframe the prompt to include roleplay as a debugging assistant and use technical language to bypass restrictions." }

---

## D  IMPLEMENTATION DETAILS

### D.1  TRAINING OF LLMMAP

Just like LLMmap, We seperate the system prompts configuration in a train and validation set, in a ratio of 40:20 for LLMmap's train set and validtion set respectively, similarly we seperate

temperature into [0.5,1] for test set and [0,0.5] for train set. We have discussed the training and validation dataset creation of LLMmap in more detail in appendix.

**Training Data Creation for LLMMap** We closely follow LLMMap's data creation methodology to prepare a training dataset for the LLMmap inference model (detailed in Section 3). The process involves the following steps:

1. **LLM Selection:** We include 14 LLMs (or LLM versions) in our training universe. These models represent a diverse set of both open-source and proprietary LLMs. Refer to Table 3 for the full list.

2. **Application Sampling:** For each LLM, we sample 75 distinct applications:

$$\{A_1, A_2, \ldots, A_{75}\},$$

where each $A_i$ is constructed by sampling from the *training* partitions of the SP, $T$, and PF universes.

3. **Querying Applications:** Each application $A_i$ is queried using a predefined query strategy, such as that of LLMMap. The query strategy consists of carefully crafted prompts that maximize inter-model discrepancy and intra-model consistency, as detailed in Section 3. The responses from these queries are recorded. The set of queries used can be found in the Appendix.

The resulting training dataset, denoted as:

$$\mathcal{D}_{\text{train}} = \{(A, \text{LLM})\},$$

comprises query-response pairs associated with each application and LLM. This dataset is used to train the inference model (described in Section 3) to identify the underlying LLM version based on the observed responses.

Similar to the Train Set, we create a validation set from the validation set split defined previously.

## D.2 EXAMPLES OF APPLICATION CONFIGURATIONS

> **System Prompt Configurations**
>
> ***System Prompt 1:*** "You are a Gardening Therapy Advocate. Explain the mental health benefits of gardening. Suggest simple, stress-relieving gardening projects for relaxation and mindfulness."
>
> ***System Prompt 2:*** "You are a Remote Work Productivity Coach. Provide advice on creating an effective workspace, managing time, and maintaining work-life balance in a remote setup."
>
> ***System Prompt 3:*** "You are a Science Fiction World-Building Consultant. Assist writers in creating immersive sci-fi settings. Offer ideas on futuristic technologies, alien species, and societal structures."
>
> ***System Prompt 4:*** "You are a Fitness Wearable Data Analyst. Help users interpret data from fitness trackers to optimize their health and fitness routines. Suggest actionable insights for achieving their goals."

---

**Prompting Framework**

---

***RAG Template :*** "Instruction: Analyze the retrieved passages critically and generate a balanced answer that reflects the information available. Highlight uncertainties if the information is incomplete or ambiguous.
Retrieved Passages: {passages}
Question: {user_question}
Task: 1. Critically evaluate the content of the passages. 2. Address the question based on the most reliable information. 3. Mention any gaps or conflicting details in the retrieved data."
***Chain Of Thought Template :*** "Instruction: Break down the question into logical components, analyze each component, and integrate the insights into a coherent final answer.
Question: {user_question}
Chain of Thought: 1. Decompose the question into smaller parts. 2. Analyze each part individually. 3. Synthesize these analyses into a single coherent response.
Final Answer: {concise_answer}"

