# OpenReview forum: "Invisible Traces: Using Hybrid Fingerprinting to identify underlying LLMs in GenAI Apps"
_ICLR.cc/2025/Workshop/BuildingTrust — Submitted to BuildingTrust_

### Official Review · Reviewer_omri · 2025-02-24

**Rating:** 5
**Confidence:** 4

**Review:**

## Summary

The paper proposes a hybrid method for fingerprinting large language models by combining targetted and un-targetted queries. For targetted queries, the paper uses a method from prior work (LLMMap), where the queries are generated by observing the outputs of various LLMs and maximizing the inter-LLM discrepancy. For untargetted queries, the method simply an uses off-the-shelf dataset of LLM interactions and trains a model to distinguish between various LLMs given their responses. The paper argues that targetted queries cannot necessarily be used in several real world scenarios, such as agentic frameworks, combinations of LLMs or dynamically changing LLMs. Hence, one needs fingerprinting methods which do not depend on dynamically generted queries.

## Strengths
* The paper addresses a timely problem of identifying LLMs deployed in a larger system, which seems to be the new dominant paradigm after chatbots
* The performance of the combined approach is well above the baselines, indicating a promising direction of future research.

## Weaknesses
### Writing.
* I don't believe that the paper is well written.
* I find the terms `static' and `dynamic' fingerprinting to be confusing. If I understand correctly, static fingerprinting constructs **targetted** fingerprints, while dynamic fingerprinting just uses model traces on **any** interaction between the user and the system. In that sense the former actually has more dynamism.
### Contributions
* I also do not fully grasp the contributions of the paper over LLMMap. In my perception, the paper claims that one does not necessarily need tailored fingerprints, and one can combine tailored and non-tailored fingerprints to get better detection? Also, the paper uses ModernBERT as opposed to the custom architecture from LLMMap to classify the models. If these are the main contributions, they do not seem to be very significant, and they are not highlighted well in the paper
* It seems like the best results are by combining static and dynamic queries, but this goes against the setting and motivation of the paper where one assumes a dynamic environment.
### Other details
* I also am not sure how the static and dynamic approaches are combined - is it an ensemble using the method from line 281 or are the static queries used for the dynamic classifier? Is one of them a better approach than the other? If the former is the case, how was $\alpha$ selected?
* Similarly, for a larger number of queries, how are the predictions combined? Is it simply majority voting?

---

### Official Review · Reviewer_kFkP · 2025-02-28
**Invisible Traces: Using Hybrid Fingerprinting to Identify Underlying LLMs in GenAI Apps**

**Rating:** 7
**Confidence:** 3

**Review:**

# Review
This paper introduces a novel hybrid fingerprinting framework that combines static and dynamic fingerprinting techniques to identify underlying Large Language Models (LLMs) in generative AI applications.


## Strengths
1. **Proposes a hybrid fingerprinting framework**, which effectively combines static and dynamic fingerprinting techniques to identify underlying LLMs in generative AI applications.
2. **Comprehensive experiments** and visualizations (e.g., t-SNE plots) demonstrate the effectiveness of the proposed method.

## Weaknesses
The framework may require retraining if the underlying models are updated or fine-tuned, which could be a limitation in dynamic environments where models evolve rapidly.

---

### Official Review · Reviewer_1oBq · 2025-03-02
**Review of paper Invisible Traces: Using Hybrid Fingerprinting to identify underlying LLMs in GenAI Apps**

**Rating:** 5
**Confidence:** 5

**Review:**

The paper categorize LLM fingerprinting into two paradigms: Static Fingerprinting and Dynamic
Fingerprinting. Then the paper presents a hybrid fingerprinting method combining these two paradigms.

Strengths:
- Convicing empirical results on the effectiveness of the proposed fingerprinting method.
- Since the dynamic fingerprinting does not require access to the original LLM, it may have broader real-world applications, especially when the LLM is unavailable.

Weaknesses and Suggestions:
- How do you set the hyperparameter $\alpha$ in Line 282? It seems that the results are heavily influenced by this hyperparameter. Does this parameter need to be carefully tuned to achieve the optimal results?
- The paper would benefit from a discussion on whether a model can still be identified from your fingerprint after fine-tuning.
- Please use `\citep{}` instead of `\cite{}` for your citations.

---

### Decision · Program_Chairs · 2025-03-04

**Decision:**

Reject

**Comment:**

This paper presents a hybrid fingerprinting framework that combines static and dynamic techniques to identify underlying LLMs in AI applications. While the topic is interesting and timely, it is not directly relevant to the workshop on Building Trust in LLMs, as it focuses more on model identification rather than transparency, alignment, or user trust. Given the low relevance to the workshop and existing methodological concerns, I recommend rejection.